# TBX1 and Basal Cell Carcinoma: Expression and Interactions with *Gli2* and *Dvl2* Signaling

**DOI:** 10.3390/ijms21020607

**Published:** 2020-01-17

**Authors:** Cinzia Caprio, Silvia Varricchio, Marchesa Bilio, Federica Feo, Rosa Ferrentino, Daniela Russo, Stefania Staibano, Daniela Alfano, Caterina Missero, Gennaro Ilardi, Antonio Baldini

**Affiliations:** 1Institute of Genetics and Biophysics, National Research Council, 80131 Naples, Italy; capriocinzia@gmail.com (C.C.); marchesa.bilio@igb.cnr.it (M.B.); rosa.ferrentino@igb.cnr.it (R.F.); daniela.alfano@igb.cnr.it (D.A.); 2Department of Advanced Biomedical Sciences, Pathology Section, University of Naples Federico II, 80131 Naples, Italy; silvia.varricchio@gmail.com (S.V.); daniela.russo@unina.it (D.R.); staibano@unina.it (S.S.); 3Department of Biology, University of Naples Federico II, and CEINGE Biotecnologie Avanzate, 80131 Naples, Italy; federica.feo@hotmail.it (F.F.); missero@ceinge.unina.it (C.M.); 4Department of Molecular Medicine and Medical Biotechnologies, University of Naples Federico II, 80131 Naples, Italy

**Keywords:** basal cell carcinoma, T-box transcription factor TBX1, genetic marker

## Abstract

Early events of basal cell carcinoma (BCC) tumorigenesis are triggered by inappropriate activation of SHH signaling, via the loss of *Patched1* (*Ptch1*) or by activating mutations of *Smoothened* (*Smo*). TBX1 is a key regulator of pharyngeal development, mainly through expression in multipotent progenitor cells of the cardiopharyngeal lineage. This transcription factor is connected to several major signaling systems, such as FGF, WNT, and SHH, and it has been linked to cell proliferation and to the regulation of cell shape and cell dynamics. Here, we show that TBX1 was expressed in all of the 51 BCC samples that we have tested, while in healthy human skin it was only expressed in the hair follicle. Signal intensity and distribution was heterogeneous among tumor samples. Experiments performed on a cellular model of mouse BCC showed that *Tbx1* is downstream to GLI2, a factor in the SHH signaling, and that, in turn, it regulates the expression of *Dvl2*, which encodes an adaptor protein that is necessary for the transduction of WNT signaling. Consistently, *Tbx1* depletion in the cellular model significantly reduced cell migration. These results suggest that TBX1 is part of a core transcription network that promotes BCC tumorigenesis.

## 1. Introduction

Basal cell carcinoma (BCC) is the most common cancer in humans [1]. Although in most cases it does not metastasize, BCC is a significant health issue because it usually appears in exposed skin, and it must be surgically removed to avoid expansion and disfiguration. Aberrant activation of Hedgehog (HH) signaling is the earliest event of BCC tumorigenesis. Cancer cells are thought to originate from epidermal stem cells of the interfollicular epidermis [2] that acquire a profile similar to embryonic hair follicle progenitor cells [3], or from the hair follicle itself [4,5,6]. Histopathologically, BCC cases are classified into three major groups, superficial, nodular, and infiltrative. In comparison to the infiltrative type, the superficial and nodular types tend to be less aggressive, grow more slowly, and have a lower risk of recurrence [7].

The TBX1 gene is haploinsufficient in DiGeorge or 22q11.2 deletion syndrome. While analyzing TBX1 target gene, we noted a gene ontology (GO) term enrichment for the “BCC pathway” [8]. This “pathway” includes genes that are involved in SHH and WNT signaling, both of which have been previously associated with *Tbx1* roles during mouse embryonic development [9,10,11]. In addition, it has been shown that *Tbx1* is downstream to SOX9 in BCC tumorigenesis [12]. In the adult mouse, *Tbx1* is expressed in the hair follicle, where it plays a role in the maintenance of the stem cell pool [13].

Here, we asked whether TBX1 is a significant marker of human BCC and we began to dissect its role in BCC cells. Results show that TBX1 was not expressed in the healthy epidermis but it was expressed in all of the BCC samples tested (*n* = 51), independently of histopathological type. However, the distribution of TBX1+ cells varied. In the cell culture model [14], GLI2 regulates *Tbx1* expression, while TBX1, in turn, regulated a key component of the WNT signaling pathway and cell migration. These data suggest that TBX1 is part of a critical pathway of BCC tumorigenesis.

## 2. Results

### 2.1. Recruitment of BCC Cases

We recruited a total of 51 cases, which included BCCs with aggressive growth, i.e., BCCs with higher risk of recurrence: Basosquamous, infiltrating, micronodular, and morphoeic (*n* = 27 or 53% of the cohort), and BCCs with indolent growth, i.e., BCCs with lower risk of recurrence: Nodular, superficial, and with adnexal differentiation (*n* = 24 or 47% of the cohort). Table 1 and Table 2 detail the composition of the cohort enrolled in this study. Anonymized samples were retrieved from the archives of the Pathology Section of the Department of Advanced Biomedical Sciences, ‘Federico II’ University of Naples. Slides of histological sections were coded and provided “blind” to the immunofluorescence (IF) analysis team who did not know the histopathological diagnosis.

### 2.2. TBX1 is Expressed in the Hair Follicle and in BCC

We performed IF using an anti-TBX1 antibody along with anti-E-cadherin (CDH1) or anti-Ki67 (MKI67) antibodies, markers of epithelial cells and proliferating cells, respectively. In healthy skin, we found TBX1 to be expressed in the hair follicle but not in the epidermis (Figure 1A,A′), while BCC cells, even when in continuity with healthy epidermis, clearly expressed TBX1 (Figure 1B,B′). All of the 51 BCC were positive for TBX1; Figure 2 and Figure 3 show examples of indolent and aggressive BCCs, respectively. In all cases, TBX1+ cells were also CDH1+, suggesting that these cells have epithelial characteristics. The distribution of TBX1+ cells within the cancer lesion was variable. We distinguished two basic patterns, polarized (P) and diffuse (D). In the P pattern, TBX1+ cells, often intensely stained, were predominantly localized to the leading edge of the BCC lesions (peripheral palisading cells) while many cells of the internal region of the tumor were TBX1-negative (see examples in Figure 2A–C and Figure 3A–C). In contrast, the D pattern was characterized by expression in most, if not all the cells of the lesion, often with a lower and uniform signal intensity (examples in Figure 2B–C′ and Figure 3B–C′). In many cases (*n* = 18, 35%), we found both patterns within the same biopsy (P + D pattern), for example Figure 2C,C′ are from the same tumor, as are the photographs in Figure 3C,C′. We also collected semi-quantitative grades of intensity of TBX1 signal where the value “1” was assigned to signal just above the background, “3” for a strong bright signal, and “2” for intermediate intensity (Table 3). In summary, TBX1 expression identifies heterogeneity between and within tumor samples.

Co-immunostaining with the cell proliferation marker Ki67 in a subset of samples, showed that in cases with “P” pattern most of the cell proliferation occurs in the peripheral, palisade layer of the BCC lesion and the majority of Ki67+ cells was also TBX1+ (76%, *n* = 404 from three samples, Figure 2A and Figure 3A).

### 2.3. TBX1 Expression and BCC Histopathology

The results described above show that TBX1 is a marker of BCC lesions but it is not a clear indicator of histopathological type of BCC because it was detected in all samples tested, regardless of the histological type. However, a closer examination of the data summarized in Table 3 suggests that some potential differences exist between the two major groups of samples. Specifically, samples that exhibited only the pattern P were predominantly of the “differentiated” group (29% vs. 18.5% of the “undifferentiated” group, Table 3). In addition, samples exhibiting a very low TBX1 signal intensity (grade “1”) were only 12.5% of the differentiated group, but were 41% of the undifferentiated group. These observations refer to a number of patients too small for formal statistical analysis, but it is tempting to speculate that more benign histological subtypes are more likely to show regular and organized TBX1 expression pattern and to present with a subpopulation of TBX1-negative cells.

### 2.4. A Murine Cellular Model of BCC Suggests a Tbx1-Dependent Pathway

To understand the role that TBX1 may have in BCC tumorigenesis pathway, we used an established keratinocyte cell line, named G2N2C, derived from a mouse model of BCC [14,15]. In this model, transgenic expression of a constitutively active mutant isoform of GLI2 in keratinocytes activates SHH signaling and induces cancer lesions similar to human BCC. *Tbx1* was expressed at high levels in G2N2C cells (Figure 4A) but it was not expressed in primary mouse keratinocytes, as expected (Figure 4D). Knock-down of *Gli2*, using a siRNA that targets the transcripts from the endogenous and the transgenic genes, led to significant downregulation of *Tbx1* expression (Figure 4B,C), indicating that expression of *Tbx1* in these cells is at least partly dependent upon GLI2/SHH signaling. Other genes of the SHH pathway, and *Sox9* were also downregulated (Appendix A). Next, we selected a set of six genes known to be regulated by TBX1 and that are related, by gene ontology enrichment, to BCC [8]. We first compared the expression of these candidate genes in normal keratinocytes and in G2N2C cells (Figure 4D). Results showed *Ptch2*, *Gli1*, *Fzd3*, and *Dvl2* to be upregulated, while *Wnt7a* was downregulated, and *Sufu* did not change significantly in G2N2C cells. Western blot analysis confirmed a high level of DVL2 protein expression in the cell line compared to primary keratinocytes (Figure 4D, bottom panel). Next, we knocked down *Tbx1* expression by siRNA and tested the expression of these genes. We found that *Dvl2* was significantly downregulated, but the expression Fzd3, Wnt5a, Wnt7a, and Sox9 was not affected (Appendix A). *Dvl2* encodes Dishevelled, an adapter protein involved in WNT signaling, and canonical and noncanonical/planar cell polarity [16]. *Dvl2* was also found to be downregulated in *Tbx1* loss of function models, including in mutant mouse embryos [8].

Overall, these results showed that SHH signaling contributes to induction or maintenance of *Tbx1* expression, and that TBX1 positively regulates the expression of a key regulator of WNT signaling pathway.

### 2.5. TBX1 is Important for Migration of G2N2C Cells

We have previously shown that, in a different context, Tbx1 interacts with the noncanonical Wnt signaling [9] and it affects cytoskeletal organization and cell movement [17]. To determine whether TBX1 affects cell migration in BCC cells we performed a scratch-wound-healing assay with G2N2C cells and monitored their response by time-lapse microscopy for 96 h. We found that TBX1 depletion substantially reduced G2N2C cell migration into the wound area (Figure 5). This effect was not due to differences in cell proliferation because cells were treated with mitomycin C. *Tbx1* is therefore a regulator of G2N2C cell migration.

## 3. Discussion

BCC, although not typically a life threatening condition, is a significant health problem because of its high prevalence, recurrence risk, and its potentially disfiguring consequences. SHH signaling is currently used as a drug target with some success [18,19], but the occurrence of resistance reduces the usefulness of this therapy. The combination of SHH and a WNT inhibitors has been shown to be efficacious in a mouse model of BCC [20]. However, there is a need for further research and the identification of additional drug targets to overcome the occurrence of drug resistance. Developmental genes are commonly involved in various types of cancer, and BCC is no exception. Activation of SHH signaling is critical in BCC initiation, and is followed by activation of WNT signaling [3]. The transcription factor SOX9, which is regulated by WNT/beta-catenin signaling [12] and by Gli transcription factors [21] orchestrates a complex transcriptional program that involves self-renewal, cytoskeletal remodeling, cell adhesion, and extra cellular matrix (ECM) regulation. Larsimont et al. [12] identified *Tbx1* as a transcriptional target of SOX9, and speculated that TBX1 might be an effector of the self-renewal functions of SOX9. Independently, we found that a group of genes targeted or regulated by TBX1 in different contexts are related to BCC pathways [8]. Here, we found that in the adult human skin, TBX1 is only expressed in the hair follicle, but it is expressed in all BCC cases tested. However, we found heterogeneity in many cases, especially in the histologically more benign cases (“indolent growth” group), where there was a substantial number of TBX1-negative cells within the lesion. The activation of the *Tbx1* gene is likely a consequence of SOX9 expression and SHH activation, as reported in the developmental context [11,22,23]. This view is supported by the finding presented here that *Gli2* suppression downregulates *Tbx1* expression in G2N2C cells. In turn, TBX1 is necessary to maintain high levels of *Dvl2* expression, an important player in the transduction of WNT signaling, canonical and noncanonical, including the planar cell polarity pathway. Indeed, loss of TBX1 was associated with reduced ability to migrate in an in vitro assay, suggesting that this transcription factor may affect cytoskeletal rearrangements as previously shown in other systems [17]. Previous work in another type of skin tumors in mice, squamous cell carcinoma (SCC), has shown that TBX1 is not expressed in these tumors and it was suggested that it might have tumor suppressor activity [24]. The differences with our data presented here may be explained by the different mechanisms of BCC tumorigenesis compared to SCC, the latter being mainly driven by RAS activation [24].

In conclusion, we show that TBX1 expression is a characteristic of BCC. As shown in Figure 6, we propose that *Tbx1* is activated by both SHH signaling and SOX9 and that downstream, TBX1 contributes to the maintenance of WNT signaling (canonical and/or noncanonical) through positive regulation of *Dvl2*. In addition, through Dvl2/PCP, or directly, TBX1 regulates dynamic characteristics of cancer cells. Further research will be necessary to establish whether in this context *Tbx1*, might trigger an autoregulatory loop (Tbx1-WNT-Sox9-Tbx1) that reduces the dependence of BCC cancer cells from SHH signaling.

## 4. Materials and Methods

### 4.1. Patients and Tissue Samples

Formalin-fixed, paraffin-embedded tissue blocks of skin biopsies of BCC cases diagnosed and excised with healthy surgical margins from September 2013 to March 2017, were retrieved from the archives of the Pathology Section of the Department of Advanced Biomedical Sciences, ‘Federico II’ University of Naples. The criteria used to select BCC cases from the archive were: The size of the biopsy (>1 cm), the availability for each case of at least two inclusions where the lesion was present, the presence of different histological subtypes of BCC (see Table 1), the documented complete excision of the lesion and the exclusion of syndromic cases. Eighty patient samples were initially selected and processed for immunofluorescence, but some had to be excluded for technical reasons (e.g., insufficient quality of immunofluorescence staining, sections lacking hair follicles, which was our internal control for staining quality, etc.). In total, we successfully tested 51 cases, 36 males and 15 females, the age at diagnosis ranged between 38 and 87 years (mean age 66.5 SD ± 13.2 years). Two pathologists reviewed the whole routine hematoxylin-eosin (H&E) sections to confirm the original diagnosis. The clinical data and pathological features of the tumors are reported in Figure 1.

The study design and procedures involving tissue samples collection and handling were performed according to the Declaration of Helsinki, in agreement with the current Italian law, and to the Institutional Ethical Committee guidelines. Informed consent was given by all patients prior to the biopsy. According to the Italian law, retrospective studies using routine archival FFPE-tissue require an informed consent prior to surgery, following the indication of Italian DLgs No. 196/03 (Privacy law), as modified by UE 2016/679 law of European parliament and Commission.

### 4.2. Tissue Microarray Construction

Out of 51 cases, 38 were provided in a tissue microarray (TMA) format. Briefly, two pathologists reviewed hematoxylin-eosin (H&E) sections to confirm the original diagnosis and to mark the most representative tumor areas useful for the TMA construction. Tissue cores with a diameter of 3 mm were punched from morphologically representative tissue areas of each ‘donor’ tissue block and brought into one recipient paraffin block using a manual tissue arrayer. The filled recipient blocks were then placed on a metal base mold. The paraffin embedding was then carried-out, by heating the blocks at 42 °C for 10 min and flattening their surface by pressing a clean glass slide on them. As a result, four TMAs were built, 4 μm sections were cut from each TMA using an ordinary microtome. The first section was stained with H&E to confirm the presence of the tumor and the integrity of tissues (examples in Appendix A). The other section was mounted on a super frost slide (Microm, Walldorf, Germany) for the immunofluorescence.

### 4.3. Immunofluorescence

Paraffin-embedded tissue blocks were cut into four μm sections and subjected to immunofluorescence, by using an anti-TBX1 antibody (Abcam, #ab18530, 1: 100), anti-E-Cadherin antibody (BD Biosciences, #610182, 1:200) anti-Ki67 antibody (BD Biosciences, #556003, 1:200), and Alexa fluor 488 (or 594) Goat anti-rabbit (or anti-mouse) IgG (H + L) (Life Technologies, 1:400) as secondary antibody. DAPI was used for nuclear staining.

### 4.4. Cell Culture Experiments

G2N2C cells were cultured under low calcium conditions in the presence of 8% Ca^2+^-chelated fetal bovine serum, and 1 ng/mL Keratinocyte growth factor (Sigma, #K1757) as described in [15] and plated in type I collagen-coated tissue culture dishes. Transient transfection of siRNA was performed by using Lipofectamine RNAiMAX reagent (Invitrogen, #13778) following the standard protocol. For reverse transfection, 1.5 × 10^6^ cells/well were plated on a 6-well plate, and 50 nM of a pool of silencer select predesigned *Tbx1* siRNA (Life Technology) and 200 nM of *Gli2* siRNA [25] was used. Nontargeting siRNA was used for control transfections. Cells were collected 24 h after transfection and processed for RNA extraction.

Total mRNA was isolated from G2N2C using 0.5 mL of QIAzol lysis reagent (Qiagen, 79306) and cDNA was synthesized from 1 µg total RNA, using the high-capacity cDNA reverse transcription kit (Applied Biosystems). Gene expression was quantified by qRT-PCR in 20 µl reaction containing 1× SYBR green (FastStart Universal SYBR Green Master (Rox), Roche) using technical triplicates. Gene expression levels were normalized to *GAPDH*. A list of primers is provided in Table 4.

For western blotting analyses, total protein extracts from mouse primary keratinocytes and from G2N2C cells were collected in a sample buffer (10% Glycerol, 0.01% Bromophenol Blue, 0.0625 M Tris-HCl pH 6.8, 3% SDS, 5% ß-mercaptoethanol) and run by gel electrophoresis. Blots were immunostained with an anti-DVL2 antibody (Elabscience E-AB-31246, 1:1000) and then with an anti- α-Tubulin antibody (SIGMA/T6074, clone B-5-1-2, 1:5000). Immunoreactivity was revealed by chemiluminescence.

For scratch wound assay, G2N2C cells were seeded in 12-well plates at a density of 4 × 10^4^ cells/well for transfection. At this density, cells reached monolayer confluency after 48 h and were treated with Mitomycin C (M4287-Sigma-Aldrich) diluted in medium (4 µg/mL) for 2 h. Wounds were performed by dragging a sterile pipette tip across the monolayer and then imaged by time-lapse microscopy. During acquisition of movies, cells were incubated in a humidified chamber at 37 °C, 5% CO2 in the complete medium and monitored for 96 h at 37 °C. A phase-contrast image was acquired in three regions along the wound every 24 h after wounding on a fully motorized DMI4000B (Leica) microscope with a Plan APO 5x and 10× objectives using LAS-AF software.

### 4.5. Statistics

All data were expressed as means ± SD from independent experiments. Statistical significance of differences was tested using the *t*-test. *p*-values < 0.05 were considered as statistically significant.

## Figures and Tables

**Figure 1 ijms-21-00607-f001:**
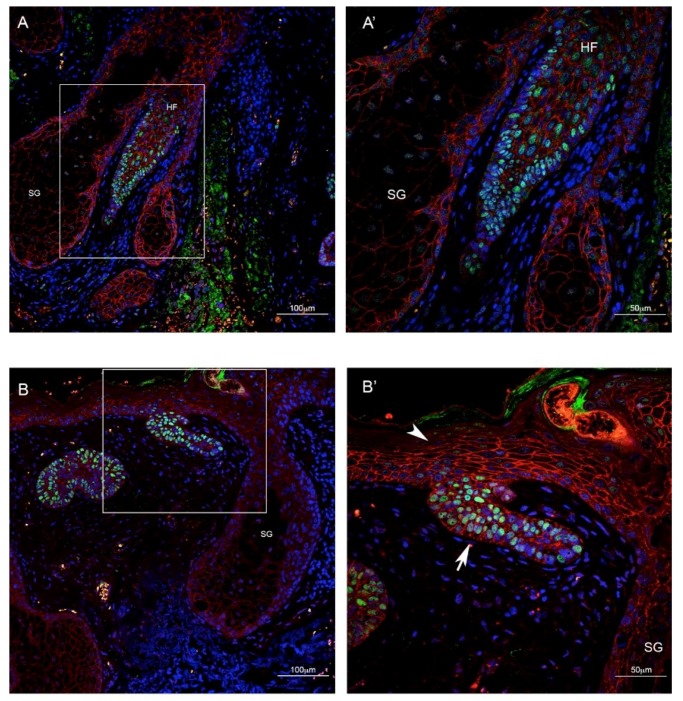
Immunofluorescence of skin biopsies from patients affected by BCC. (**A**) TBX1 (green), E-cadherin, an epithelial cell marker (red), and 4′,6-diamidino-2-phenylindole (DAPI), which stains DNA (blue) showing the physiological expression of TBX1 in the hair follicle (HF, magnified in **A′**). (**B**) BCC connected to the epidermal layer: TBX1 is expressed in cancer cells (arrow, magnified in (**B′**) but not in healthy epidermis (arrowhead). SG: Sebaceous gland. Scale bar is 100 µm in (**A**,**B**), 50 µm in (**A′**,**B′**).

**Figure 2 ijms-21-00607-f002:**
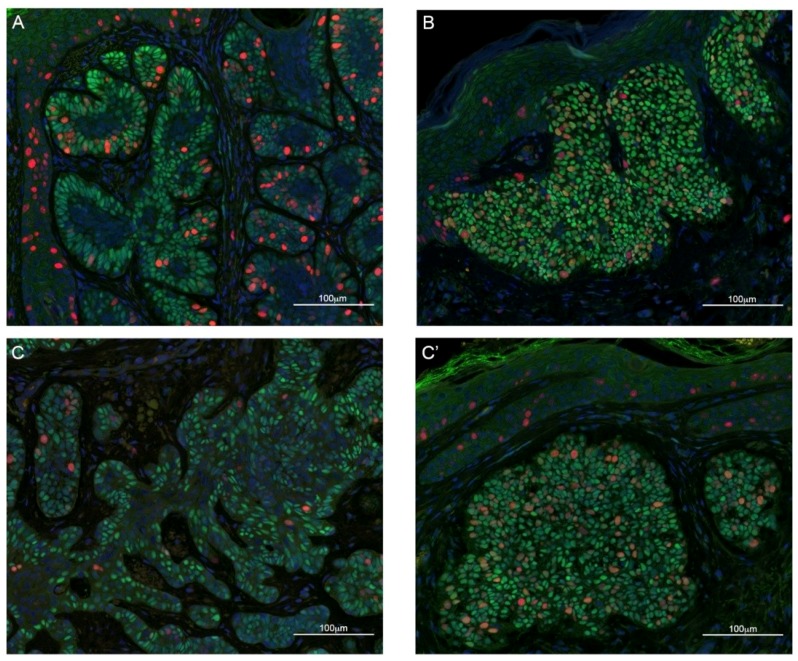
Examples of immunofluorescence staining of sections from skin biopsies of BCC with indolent growth. TBX1 in green, Ki67 (a cell proliferation marker) in red, and DAPI (DNA stain) in blue. (**A**) Example of polarized distribution of TBX1+ cells, defined as localization of positive cells at the periphery of the lesion (palisade cells), while internal cells are mostly TBX1-negative; (**B**) example of diffuse distribution, defined as localization of positive cells to the entire lesion. (**C**,**C′**) represent different regions of the same sample; note that both types of distribution, polarized (**C**), and diffuse (**C′**) are present. Scale bar 100 µm. See Appendix A for examples of hematoxylin-eosin staining of these biopsies.

**Figure 3 ijms-21-00607-f003:**
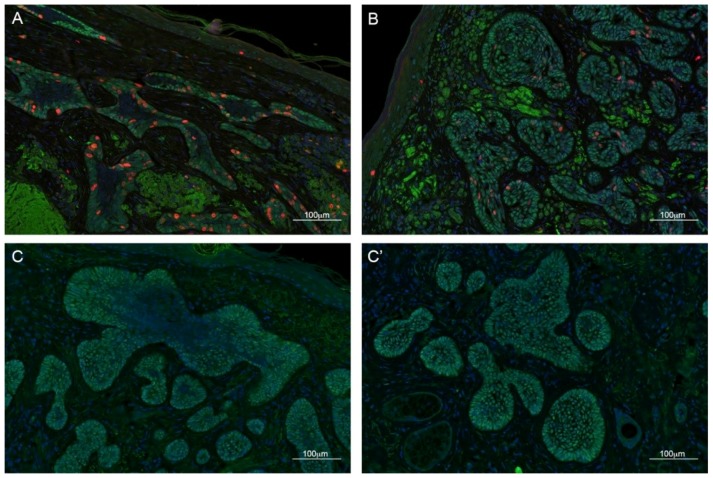
Immunofluorescence examples of section from BCC cases with aggressive growth. TBX1 in green, Ki67 (a marker of cell proliferation), in red, and DAPI in blue. (**A**) Case with a polarized distribution (i.e., mostly limited to the periphery of BCC) and (**B**) with a diffuse distribution. (**C**,**C′**), two regions of the same tumor with different distribution of TBX1+ cells. Scale bar is 100 µm. See Appendix A for examples of hematoxylin-eosin staining of these biopsies.

**Figure 4 ijms-21-00607-f004:**
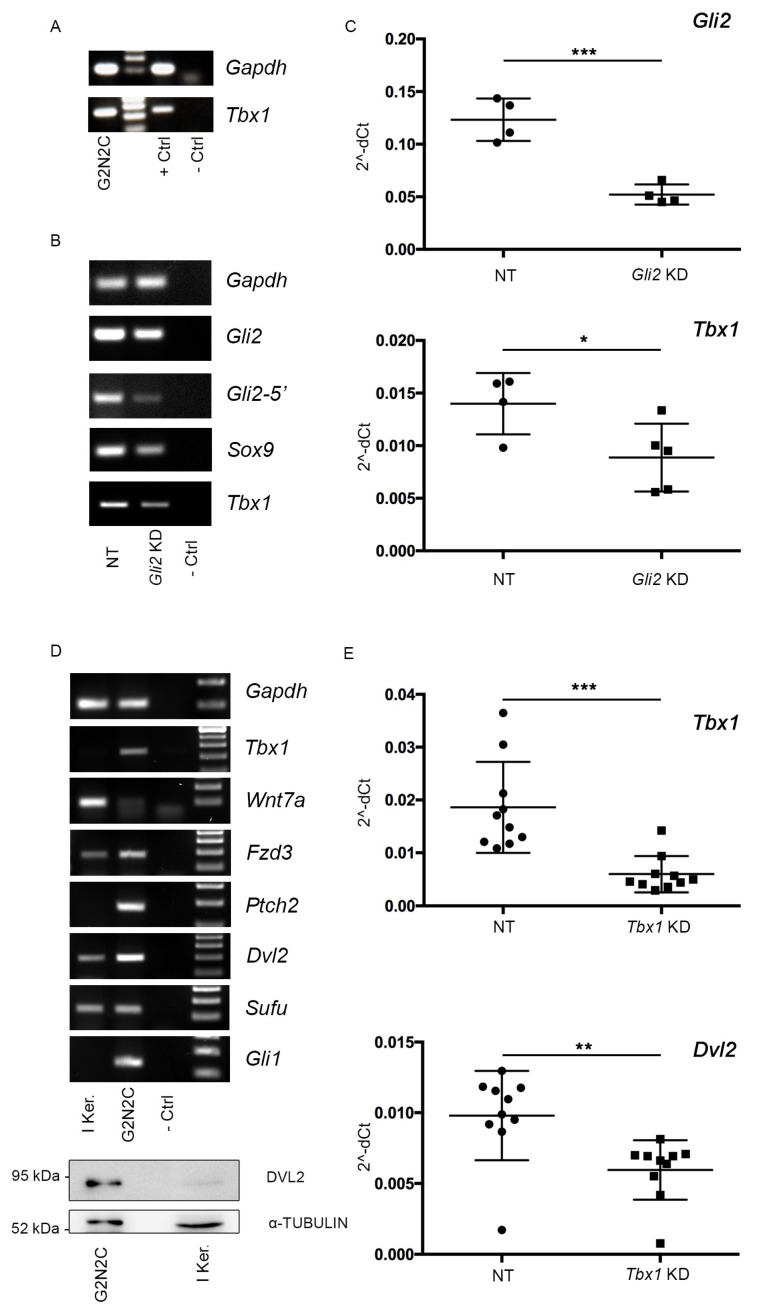
(**A**) Tbx1 expression in G2N2C cells assayed by reverse transcription (RT)-PCR. Positive and negative controls (+ Ctrl and – Ctrl) are cDNA from an E9.5 mouse embryo and from an RT sample, respectively. (**B**) Assay of the genes indicated in nontargeted (NT) and *Gli2*-trageted short interfering RNA in G2N2C cells. (**C**) Summary of expression of *Gli2* (top panel) and *Tbx1* assayed by quantitative real time qRT-PCR in multiple experiments of *Gli2* knock down in G2N2C cells. Each dot represents an independent experiment. (**D**) Expression of the genes indicated in primary keratinocytes and in G2N2C cells. Bottom panel: Western blotting analysis of total extracts from G2N2C cells and primary keratinocytes immunostained with an anti DVL2 antibody and with an anti-beta tubulin as loading control. (**E**) Summary of qRT-PCR experiments after *Tbx1* knock down. The top panel shows the expression of *Tbx1*, the bottom panel shows the expression of *Dvl2*. * *p* < 0.05; ** *p* < 0.005; *** *p* < 0.001.

**Figure 5 ijms-21-00607-f005:**
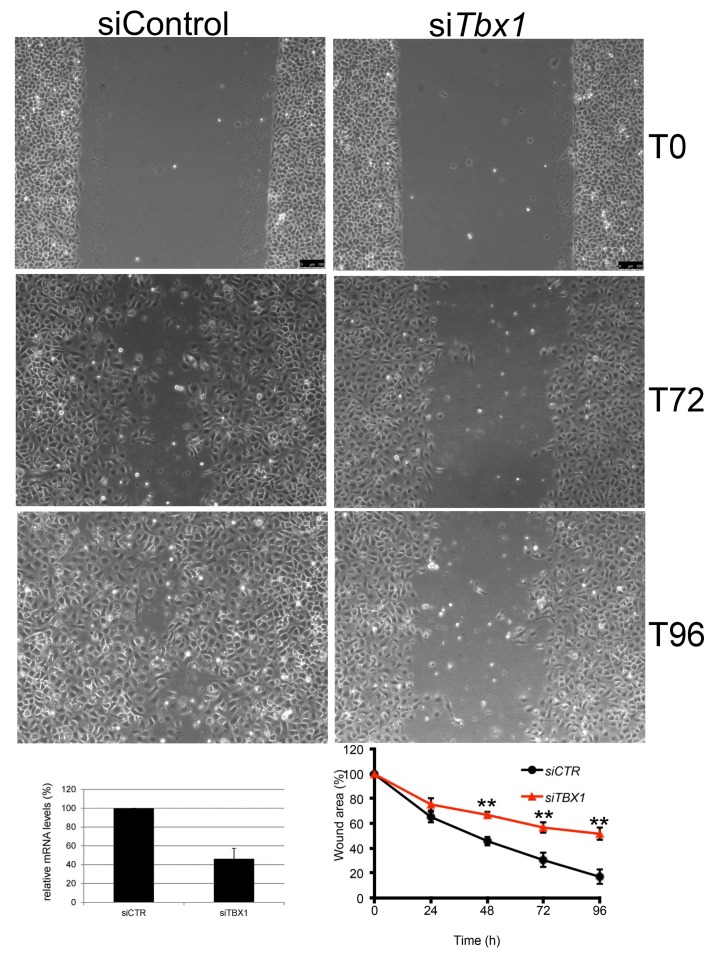
Wound healing assay with G2N2C cells. The phase contrast photographs show the wound at T0, T72, and T96 h in control (siControl) and *Tbx1* knock down (*siTbx1*) cells (scale bar: 100 µm). The histogram on bottom left shows the average of *Tbx1* expression in four independent experiments used for the assay (individual numbers were 57%, 34%, 54%, and 40% of the control expression). The graph on the bottom right panel shows the scratch wound area as a percentage of the initial wound area determined using ImageJ software. The values are the means ± SD of four independent experiments (** *p* < 0.01). *p*-values for 48, 72, and 96 h are 0.0013, 0.006, and 0.0016, respectively.

**Figure 6 ijms-21-00607-f006:**
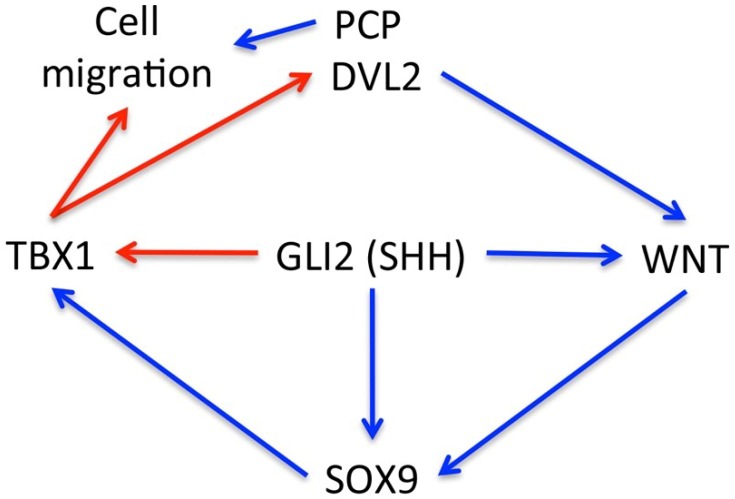
Schematic of the proposed regulatory role of TBX1 in BCC. Interactions already established are shown in blue [12,16,21], those proposed here are shown in red.

**Table 1 ijms-21-00607-t001:** Histological diagnosis of samples included in the study.

Histological Subtype	*n*
**BCC with aggressive growth**	**27**
Basosquamous Carcinoma	2
Infiltrating BCC	17
Micronodular BCC	2
Sclerosing/morphoeic BCC	6
**BCC with indolent growth**	**24**
BCC with adnexal differentiation	5
Nodular BCC	15
Superficial BCC	4
**Total**	**51**

**Table 2 ijms-21-00607-t002:** Clinicopathological characteristics of the study population.

**Age**	Mean ± SD	66.5 ± 13.2 years	
	Range (Min–Max)	38–87 years	
**Sex**	Male	*n* = 36	71%
	Female	*n* = 15	29%
**Tumor Site**	Area H	*n* = 18	35%
	Area M	*n* = 6	12%
	Area L	*n* = 24	47%
	ND	*n* = 3	6%
**Histologic Subtype**	BCC with indolent growth	*n* = 24	47%
	BCC with aggressive growth	*n* = 27	53%
**Tumor Size**	>2 cm	*n* = 7	14%
	<2 cm	*n* =36	71%
	ND	*n* = 8	16%

*n* = 51; Area H: “mask areas” of face (central face, eyelids, eyebrows, periorbital, nose, lips [cutaneous and vermilion], chin, mandible, preauricular and postauricular skin/sulci, temple, ear), genitalia, hands, and feet; Area M: Cheeks, forehead, scalp, neck, and pretibial; Area L: Trunk and extremities.

**Table 3 ijms-21-00607-t003:** Results of TBX1 immunofluorescence assay.

Sample Code	Histological Type	Differentiation Sstate	TBX1+ Cells	TBX1 Signal
15	Nodular BCC	differentiated	D	2
BAS59	Nodular BCC	differentiated	D + P	3
BAS60	BCC with adnexal differentiation	differentiated	D + P	3
BAS5	Nodular BCC	differentiated	D	3
BAS11	Nodular BCC	differentiated	P	2
BAS14	Superficial BCC	differentiated	D	3
BAS24	Nodular BCC	differentiated	D + P	3
BAS29	Superficial BCC	differentiated	D	2
BAS38	Nodular BCC	differentiated	P	1
BAS40	Nodular BCC	differentiated	P	1
BAS41	Nodular BCC	differentiated	P + D	2
BAS42	Nodular BCC	differentiated	D	3
A	BCC with adnexal differentiation	differentiated	D	1
8	Nodular BCC	differentiated	D	2
2	Nodular BCC	differentiated	D + P	3
B	Superficial BCC	differentiated	P	2
E	Superficial BCC	differentiated	D	3
G	BCC with adnexal differentiation	differentiated	P	2
I	Nodular BCC	differentiated	P	2
L	BCC with adnexal differentiation	differentiated	P	2
M	Nodular BCC	differentiated	D + P	3
n	Nodular BCC	differentiated	P + D	2
R	Nodular BCC	differentiated	D	3
S	BCC with adnexal differentiation	differentiated	D + P	3
BAS7	Micronodular BCC	undifferentiated	P	2
BAS8	Infiltrating BCC	undifferentiated	D	1
BAS9	Basosquamous Carcinoma	undifferentiated	P	1
BAS12	Infiltrating BCC	undifferentiated	P + D	2
BAS13	Infiltrating BCC	undifferentiated	D	2
BAS19	Infiltrating BCC	undifferentiated	D	3
BAS67	Sclerosing/morphoeic BCC	undifferentiated	D	1
BAS68	Micronodular BCC	undifferentiated	D	3
BAS69	Infiltrating BCC	undifferentiated	P + D	1
BAS28	Infiltrating BCC	undifferentiated	D	1
BAS44	Basosquamous Carcinoma	undifferentiated	D	2
BAS47	Sclerosing/morphoeic BCC	undifferentiated	P + D	2
BAS48	Infiltrating BCC	undifferentiated	P + D	2
BAS51	Infiltrating BCC	undifferentiated	D + P	3
12	Infiltrating BCC	undifferentiated	D	2
3	Infiltrating BCC	undifferentiated	D	2
1	Infiltrating BCC	undifferentiated	P + D	2
C	Sclerosing/morphoeic BCC	undifferentiated	D + P	1
BAS81	Infiltrating BCC	undifferentiated	D + P	1
D	Sclerosing/morphoeic BCC	undifferentiated	D	1
F	Sclerosing/morphoeic BCC	undifferentiated	D	1
H	Sclerosing/morphoeic BCC	undifferentiated	D	1
O	Infiltrating BCC	undifferentiated	P + D	3
P	Infiltrating BCC	undifferentiated	P + D	3
Q	Infiltrating BCC	undifferentiated	P	2
T	Infiltrating BCC	undifferentiated	P	3
BAS89	Infiltrating BCC	undifferentiated	P	1

Distribution of TBX1+ cells and score of the intensity of the immunofluorescence signal (1: Weak above background, 3: Clear, strong signal, 2: Intermediate intensity) for each case studied. P: Polarized; D: Diffuse. See text for definitions.

**Table 4 ijms-21-00607-t004:** Sequence of primer pairs used in this study.

Primer Name	Sequence 5′-3′
*Gapdh*-F:	TGCACCACCAACTGCTTAGC
*Gapdh*-R:	TCTTCTGGGTGGCAGTGATG
*Tbx1_qRT-PCR*-F:	CTGACCAATAACCTGCTGGATGA
*Tbx1_qRT-PCR*-R:	GGCTGATATCTGTGCATGGAGTT
*Tbx1_RT-PCR*-F:	TTTGTGCCCGTAGATGACAA
*Tbx1_RT-PCR-R*:	AATCGGGGCTGATATCTGTG
*Gli2*-F:	ATCAAGACAGAGAGCTCCGG
*Gli2*-R:	ATGCCACTGTCATTGTTGGG
*Gli2-5′*-F:	GCCGATTGACATGAGACACC
*Gli2-5′*-R:	CTGAAGGGTGACTCTCCAGG
*Sox9*-F:	GTACCCGCATCTGCACAAC
*Sox9*-R:	CTCCTCCACGAAGGGTCTCT
*Wnt7a*-F:	CGCTCATGAACTTACACAATAACG
*Wnt7a*-R:	ACAGGAGCCTGACACACCAT
*Fzd3*-F:	AGGTGGGCACAGTTTGTTTT
*Fzd3*-R:	GAAATGGCCGAAAATCCCGA
*Ptch2*-F:	GTCCACCTAGTGCTCCCAAC
*Ptch2*-R:	GTGCCCCCTAGTAGCAGTTC
*Dvl2*-F:	AGCAGTGCCTCCCGCCTCCTCA
*Dvl2*-R:	CCCATCACCACGCTCGTTACTTTG
*Sufu*-F:	TACTACGGACAGTGCCCATT
*Sufu*-R:	GTCTGTCTCAATGCCTTTGTCA
*Gli1*-F:	GAATTCGTGTGCCATTG
*Gli1*-R:	GGACTTCCGACAGCCTTCAA

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
