# Peer review of "TBX1 and Basal Cell Carcinoma: Expression and Interactions with Gli2 and Dvl2 Signaling"

_ijms, 2020, doi:10.3390/ijms21020607_

Round 1

We thank the reviewers for the thorough reading of our manuscript and constructive comments. To address the reviewer's comments, we have added 3 new figures and added more details in two of the original figures. Text was also modified.

Following is our point-by-point response to comments.

This research is well performed. I consider the article suitable for the publication in IJMS with minor revisions.

1) Figure 1 shows the expression of TBX1 in BCC sample patients by immunofluorescence analysis. The author should report also the corresponding quantitative data (histograms).

Response:

Immunofluorescence is a semi quantitative method; we do not have histograms for it. Signal intensity was evaluated semi quantitatively on a scale 1 to 3 (Table 3), where 1 is just above background, 3 is a strong bright signal, and 2 is in between.

2) The expression of TBX1, Ptch2, Gli1, Fzd3, Wnt7a and Dvl2 in murine cellular model of BCC was analysed through RT-PCR. In my opinion, the author must analyse also the corresponding protein level. mRNA level variations not always lead to variations in protein levels.

Response:

We include a western blot for DVL2 (Fig. 4D); unfortunately, we do not have good antibodies for the other proteins. We have added quantitative rt-PCR for the other genes (Fig. S2 and S3).

Round 2

Cinzia Caprio and co-authors have deeply revised the manuscript addressing most of the reviewer's comments. This revised manuscript is suitable for the publication in IJMS.

Author Response

Thank you for the positive response.

Round 1

We thank the reviewers for the thorough reading of our manuscript and constructive comments. To address the reviewer's comments, we have added 3 new figures and added more details in two of the original figures. Text was also modified.

Following is our point-by-point response to comments.

1) The figure legends lack sufficient detail.

Figure 1 shows the expression of TBX1 in skin biopsy samples although it does not state this or give details about where the samples came from. The hair follicle is indicated in panel A, and a BCC lesion in panel B. The two panels are not directly comparable as no hair follicle is shown in B. There are also patches of green fluorescence in both samples which are not mentioned. There are arrows in panel B but no description as to what they are pointing to. Presumably the proximity of the hair follicle to a sebaceous gland is sufficient to identify it, but this is not explained. Nor is what E-cadherin is staining. This is assuming the reader will have sufficient knowledge, but it is better to explain these things for the non-expert.

Response:

                  We have checked, updated, and extended all the Figure legends, as suggested.

2) In the materials and methods it is explained that H&E sections were used to confirm the BCC diagnosis. It would be interesting to see examples of this in the paper.

Response:

We have added H&E microphotographs related to samples shown in Figs. 2, and 3 (new Fig. S1)

3) Figure 2. It is not clear how “polarized” and “diffuse” phenotypes are defined. Although this is described in the main text there is not enough detail in the legend to determine this and the figures do not clearly show details at high power. The main text mentions that all TBX1 positive cells are CDH1 positive, but this staining is not shown in the figure. Ki67 staining is shown but not commented on.

Response:

                  We have defined the two patterns in the figure legend, too. CDH1 staining is shown in Fig. 1. We now state that Ki67 is a marker of cell proliferation.

4) Figure 3. Same comments as for Figure 2. In this figure the scale bar representing 100um is similar in size to the one shown in Fig. 2 but the images of the tissue looks to be at much lower magnification. Please confirm the accuracy of the scale bars.

Response:

                  Indeed, photographs of the two figures were printed at different magnification in an effort to use space more efficiently. The scale bars are also of different size in the two figures, proportional to the magnification, and are accurate.

5) Line 106-108

                  The Ki67 staining is mentioned but there is not enough detailed analysis to understand what this means. What is meant by “…a subset of samples with the “P” pattern…”? Why is only a subset of the “P” pattern selected for Ki67 analysis? What is the proliferative status of the “D” cells? Much more could be made from analysing the Ki67 staining, with graphs and statistical analysis. A value of 67% is given but no indication to the number of samples analysed or standard deviation. Can control samples be included in the analysis for comparison? Do TBX1 positive cells in the hair follicle have Ki67 staining?

Response:

                  The value of 76% is derived from counting a total of 404 Ki67+ cells from 3 samples with P pattern. We provide this information in the revision. We performed Ki67 staining in most samples, but we comment only on the P pattern because in the D pattern all or nearly all Ki67+ cells are TBX1+ simply because most if not all cells of the BCC are TBX1+. We do not draw any conclusion from Ki67 and did not intend to make any comparison. The point was simply to show that TBX1+ cells are in the periphery of tumors (palised cells) where proliferating cells are also located.

The TBX1+ cells in the hair follicle are also proliferating, but we have not scored the proliferation rate.

6) Line 114

Are the numbers for staining pattern and differentiation state significantly different? Can this data be tested to see if the differences are meaningful?

Response:

                  In the revision we state: "These observations refer to a number of patients too small for formal statistical analysis, but it is tempting to speculate that more benign histological subtypes are more likely to show regular and organized TBX1 expression pattern and to present with a subpopulation of TBX1-negative cells."

7) Line 117

Further analysis is required as the statements made are too vague (e.g. “common” versus “uncommon”, “relatively small and qualitative”, “trend”, etc.). These terms need to be backed up with some more robust analysis so more definitive statements can be made about the pathology and TBX1 expression.

Response:

                  Please see our response to point 6 above.

8) Line 128.

siRNA was used to knockdown Gli2, and this targeted “the endogenous WT and the mutant isoform”. What is the mutant isoform? Is it different to the wild-type? Please clarify.

Response:

                  The G2N2C cell line was derived from mice that carry a transgene expressing a mutant isoform of Gli2. These cells also have two copies of the endogenous gene Gli2. Our siRNA targets RNA transcripts form the transgenic and endogenous genes. This is clarified in the Results.

9) Figure 4. More detail is required in the legend so these experiments can be understood. E.g. are all the experiments shown performed in G2N2C cells?

Response:

                  We added details in the legend. Yes, all in vitro experiments were performed with the same cell line, except that in panel D of Fig. 4 RNA (and protein extracts) were also obtained from primary keratinocytes for comparison.

10) It is not clear if the samples shown for Gli2 in panel B are the same as in panel C. With respect to the RT-PCR graphs in panels B and D, why is only an end-point PCR used to ascertain gene expression in control and experimental cells when a quantitative method such as qPCR could be employed? This would surely be a much better way to assess changes in gene expression following knockdown or the differences between different cell types.

Response:

                  Panel B shows data from a single experiment (shown as an example). Panel C summarize results from multiple independent experiments (each dot refers to an independent, biological replicate), including the one shown in Panel B. The new Fig. S2 shows real time PCR for genes Sox9, Gli1, and Ptch2, which are also down regulated in Gli2 knock down experiments.

11) The graphs have unlabelled Y axes. KD is not defined. Are the qPCR experiments performed in triplicate for each sample?

Response:

                  This has been corrected and specified. All qRT tests are performed with technical triplicates, as well as biological replicates. Methods and figure legends are now clear about these points.

12) These experiments and the results obtained are the basis for the claims that TBX1 is downstream of SHH and interacts with WNT signalling. But quantitatively examining only one representative gene per pathway is insufficient to be able to draw these conclusions. These experiments should therefore be expanded on with analysis of further representative genes to be confident that these pathways are robustly shown to be involved with TBX1.

Response:

                  Indeed, we have tested a limited number of genes for each pathway, and we have observed significant differences with Gli2 (upstream) and Dvl2 (downstream). The new Fig S3 confirms by qRT-PCR that Tbx1 knock down does not affect the expression of Fzd3, Wnt7a, Wnt5a, and Sox9. Therefore, in the revision we indicate gene names rather than pathway names, when appropriate. Please note that it has already been sown that SHH regulates (indirectly) Tbx1 expression and that Tbx1 interacts with WNT in vivo, in a developmental context, but molecular details are not defined yet. The appropriate references are cited in the manuscript.

13) Figure 5. This figure only shows the final time-point of the scratch wound assay. It would be more informative to show more pictures throughout the time course (e.g. 0, 48, 96) to get a better feel of the differences between the two cell populations. Has the level of Tbx1 KD been assessed in these cells? Is there an equivalent KD of Tbx1 in each of the four independent experiments?

P values need to be indicated on the graph.

Response:

                  The revised Figure 5 includes T0, T72, and T96, as well as KD values in replicates, which are similar. P-value asterisks are now indicated in the graph.

14) Table 4.

Incorrect title used for the table.

Response:

Corrected in the revision.

15) Discussion

The authors mention that Tbx1 activation is likely to be a consequence of SOX9 expression and SHH activation. Can the activity of SOX9 also be shown in the BCC samples/cell model to demonstrate this?

Response:

The activation of Tbx1 by Sox9 has already been published (Larsimont et al., 2015). We have added in the revision the expression of Sox9 in response to Gli2 and Tbx1 depletion in the cell line (Figs. S2 and 3). Sox9 is down regulated by Gli2 knock down but not by Tbx1 knock down.

16) BCC is described as a non-metastasizing cancer. The data shown suggest that Tbx1 expression drives cell migration. Can the authors comment on how increased TBX1 expression in BCC does not lead to metastasis?

Response:

                  Our assay tests migration rather than invasiveness. The latter, more directly linked to the generation of metastasis, is hard to assay in vitro because it depends on many factors (e.g. tumor microenvironment). Our migration test is consistent with the involvement of Dvl2, which is also linked to cell migration.

17) Throughout the manuscript there are spelling mistakes and grammatical errors that need correcting.

Response:

                  The revised manuscript has been thoroughly checked by a mother-tongue English professional.

Round 2

The authors have responded to all my comments to my satisfaction.

Fig S2, however, needs to have the statistically significant differences indicated on the graph.

Author Response

Thank you for the positive response. We have added the P values on the graph of a revised version of Figure S2.

Round 1

We thank the reviewers for the thorough reading of our manuscript and constructive comments. To address the reviewer's comments, we have added 3 new figures and added more details in two of the original figures. Text was also modified.

Following is our point-by-point response to comments.

The manuscript (MS) aims to evaluate if TBX1 is a significant marker of human BCC. The study is well thought, and its findings are interesting. I still have some comments which need to be addressed. Some minor English editing is also necessary.

1) Line 35 – Replace the reference “(Epstein 2008)” with its number. Line 48 – Replace “BCC lesion” with “BCC”. Line 50 – Replace “independently from” with “independently of the”.

Response:

The text has been edited as suggested.

2) Materials and methods

Line 210 – The authors should explain the method used to retrieve only 51 BCCs cases from the archives, considering the period of September 2013 to March 2017. How many BCC cases were there in the archives in this period? It is not clear how these 51 cases were selected. Line 212 – The 51 BCC cases belonged to 51 patients. None of these patients had more than one BCC? Line 215 – “Fig. 1” should be “Table 1”. Line 220 – “Some of the samples…” The authors should be more specific and state how many cases were provided in TMA.

Response:

We added in the revision: "Criteria for recruitment of cases in this study were: size of biopsy (>1cm), histological types. 80 patients samples were initially selected and processed for immunofluorescence, but some had to be excluded for technical reasons (e.g. insufficient quality of immunofluorescence staining, sections lacking hair follicles, which was our internal control for staining quality, etc.). "

We also specify that 38 out of 51 cases were provided in TMA format.

3) Line 60 – The sentence “Anonymized…Naples” is repeated in the Materials and methods Section, so I suggest to remove it from the Results Section. Line 114 – “(29% vs. 18.5%...) – This difference was not statistically significant, right? The authors should mention that. The same happens in the next sentence “…(41%).” Referring to the previous comment, if these differences are not statistically significant (p-values are not mentioned) what do the authors mean with “the trend is that the more benign…”? This seems an overstatement. Line 122 - Replace “in a BCC tumorigenesis pathway” with “in the BCC tumorigenesis pathway”.

Response:

The sentence about the anonymized samples is now only present in the Results section, as we think that it is important to state the workflow at that point.

Concerning the relationship between TBX1 expression and histopathology, indeed the numbers are too small for statistical analyses. In the revision we state: "These observations refer to a number of patients too small for formal statistical analysis, but it is tempting to speculate that more benign histological subtypes are more likely to show regular and organized TBX1 expression pattern and to present with a subpopulation of TBX1-negative cells."

4) Discussion

Line 176 – “In the same work…” It is not clear what work the authors are referring to. Please, clarify. Line 180 - Replace “but is also expressed” with “but it is expressed”. Line 183 – The authors state: “It is tempting to speculate that the presence of TBX1-negative … No data were shown to support this speculation, so I believe this statement should be removed. Line 191 – Replace “Previous work in other types” with “Previous work in another type”. Line 197 - Replace “cartoon” with a more adequate word such as “diagram”, or “illustration”, or “scheme”.

Response:

The text has been edited as suggested.

5) Tables

Table 2 – Include standard deviation in the row of mean age. Table 3 – The title is not correct. On the footnote, the description for immunofluorescence signal 2 is missing. Table 4 – The title is not correct.

Response:

We have edited the text and tables as suggested.

6) Figure 4 – Asterisks appear in the graphs but their meaning does not appear in the caption. Figure 5 – Contrarily to the previous figure, asterisks appear in the caption but they do not appear in the graph.

Response:

This has been corrected.

Round 2

The authors made an effort to fully improve the manuscript (MS). There is only one comment which was not fully addressed.

The authors explained that they retrieved the 51 BCCs cases analyzed from 80 initial cases with the criteria:  size of biopsy (>1cm), and histological types (which ones ? Please define). I still have a doubt: were these 80 initial BCC cases sequential, meaning that considering the period of September 2013 to March 2017, there were only 80 cases in the archives  which met the selection criteria the authors previously defined? Please, clarify.

Author Response

Response:

We are sorry that our response to the original comment was not satisfactory. Indeed, the description of criteria was incomplete. Thank you for pointing it out. We have revised the description as follows: "The criteria used to select BCCs cases from the archive were: the size of the biopsy (> 1cm), the availability for each case of at least two inclusions where the lesion was present, the presence of different histological subtypes of BCC (see Tab. 1), the documented complete excision of the lesion and the exclusion of syndromic cases."